# Exploring the Connection between the TDD Practice and Test Smells—A Systematic Literature Review †

Matheus Marabesi [1,*] , Alicia García-Holgado [2,*] and Francisco José García-Peñalvo [3,*]

GRIAL Research Group, Universidad de Salamanca, 37008 Salamanca, Spain
* Correspondence: matheus.marabesi@gmail.com (M.M.); aliciagh@usal.es (A.G.-H.); fgarcia@usal.es (F.J.G.-P.)
† This paper is an extended version of our paper published in the 18th Iberian Conference on Information Systems and Technologies (CISTI'2023), Aveiro, Portugal, 20–23 June 2023.

**Abstract:** Test-driven development (TDD) is an agile practice of writing test code before production code, following three stages: red, green, and refactor. In the red stage, the test code is written; in the green stage, the minimum code necessary to make the test pass is implemented, and in the refactor stage, improvements are made to the code. This practice is widespread across the industry, and various studies have been conducted to understand its benefits and impacts on the software development process. Despite its popularity, TDD studies often focus on the technical aspects of the practice, such as the external/internal quality of the code, productivity, test smells, and code comprehension, rather than the context in which it is practiced. In this paper, we present a systematic literature review using Scopus, Web of Science, and Google Scholar that focuses on the TDD practice and the influences that lead to the introduction of test smells/anti-patterns in the test code. The findings suggest that organizational structure influences the testing strategy. Additionally, there is a tendency to use test smells and TDD anti-patterns interchangeably, and test smells negatively impact code comprehension. Furthermore, TDD styles and the relationship between TDD practice and the generation of test smells are frequently overlooked in the literature.

**Keywords:** TDD; test smells; anti-patterns; agile; practitioners; software development; systematic literature review

## 1. Introduction

In an agile environment, the normalization of responses to change should be prioritized, with the code accompanying and keeping pace with evolving business requirements. In that sense, the test code is utilized as a safety net to evolve and prevent regressions from occurring. However, the challenges faced by practitioners, as research suggests, remain constant. Code comprehension, for example, should be treated as a priority, as it indicates that developers spend most of their time understanding code rather than writing it from scratch [1]. The test code that has test smells might suffer from the same cause. Testing code is also used by practitioners to understand what the application does as a living document, turning it into a critical path for continuous improvement.

Test-driven development (TDD), as it is known, has changed how practitioners think about developing software entirely. The practice was popularized by eXtreme Programming, one of the earliest methodologies to influence agile practices, which prioritize code testability at the core of software development to support business dynamics. The practice of writing the test code before the production code follows three stages [2]:

- Red—Write the test code first, thinking about how it should behave.
- Green—Make the test pass with the least possible number of changes.
- Blue—Refactor, remove any duplicate code, and make quality improvements.

Following these stages, the promise is that it will also enhance code quality and mitigate code smells. Researchers have challenged this idea, and through different studies,

according to Ghafari et al. [3], the findings are inconclusive. Therefore, Aniche [4] offered empirical evidence that developers value the TDD cycle and its feedback. The Section 2 of this work also corroborates with the inconclusive results regarding productivity. However, it remains a practice that is popular among practitioners. Such popularity has led variations in the practice. Two of the most popular ones are the Chicago School (known as inside-out) and the London School (known as outside-in) [5].

On one hand, inside-out focuses on the business stages first and the interactions that objects should have to fulfill a given requirement. This style serves as a learning path in the book 'TDD by Example' by Beck [2]. The book uses a calculator as a fictional business requirement to test drive the implementation and guide the learner step-by-step.

On the other hand, outside-in focuses on the opposite. It starts with the client of the application, such as an API (application programming interface) or a graphical user interface, and then delves into the inside where the business stages are written and will be fulfilled. This style is also referred to as TDD double-loop.

Freeman and Pryce used an auction system to illustrate the usage of this style with two levels of feedback [6]. Sandro Mancuso and Robert Martin created a comparison video series regarding both styles: London vs. Chicago—Introduction: https://www.youtube.com/watch?v=v68osKXat90 (accessed on 11 January 2024).

The outside-in style is closely related to the history of test doubles and how they were created. Meszaros [7] also connects the usage of test doubles to this style. What is known today as mock frameworks and different types of test doubles were popularized there when practitioners faced challenges testing their applications with real implementations [8], leading to studies being published on how to properly work with test doubles [9] and how developers use mock objects [10].

Regardless of the adopted style, systematic literature reviews over the past 13 years have predominantly focused on aspects of TDD related to TDD practice, TDD quality, and TDD productivity. These are indeed core areas for practitioners to adopt it. However, the TDD practice in the literature is found to be isolated from the context in which it is applied; few studies mention the team settings in which TDD is practiced and the link between the practice and the generation of test smells.

Research has shown that test smells are widely covered from various angles. The community has provided empirical evidence that test smells are harmful for comprehension and how they spread in both open-source and industrial projects. Nevertheless, practitioners rank them as a low to medium priority, despite their negative impact on code comprehension. Therefore, tools were developed and empirically evaluated to address the need for improving test smells in test suites.

Through a systematic literature review, this study explored TDD from a broader perspective, focusing on the aspects that practitioners use it, team settings, and TDD styles that might lead to the generation of test smells. The results suggest that when creating the testing strategy, various aspects of the organization are taken into account, such as roles, desired outcomes, balancing the testing strategy with delivery, and the level of maturity. When TDD comes into play, practitioners report that due to project pressure, the practice of TDD might be impacted and even dropped.

The remainder of this work is organized as follows: Section 2 provides an overview of the related studies preceding this systematic review. Section 3 outlines the methodology employed in this study. Section 4 presents and discusses the findings obtained. Subsequently, in Section 5, the implications of these findings are analyzed. Finally, Section 6 summarizes the study and offers concluding remarks.

## 2. Related Work

The literature focused on systematic literature reviews over the past 13 years has shown that TDD is a subject that has been widely researched, ranging from challenges in TDD adoption to its effectiveness and guidelines for practitioners. This section covers the systematic literature reviews that have already been conducted, which laid the groundwork

for the need to conduct this study. Figure 1 depicts the studies across publishers and shows that IEEE is the publisher that holds the largest number of systematic literature review for TDD, followed by Elsevier.

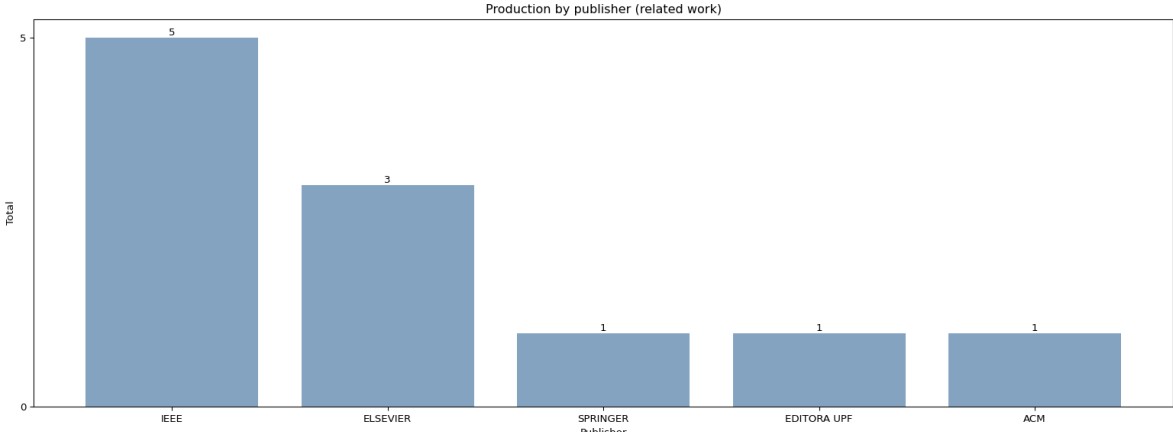

**Figure 1.** Related systematic literature reviews distributed by publishers.

Figure 2 depicts the related work evenly distributed across the years (one each) except for 2016, which had two published studies.

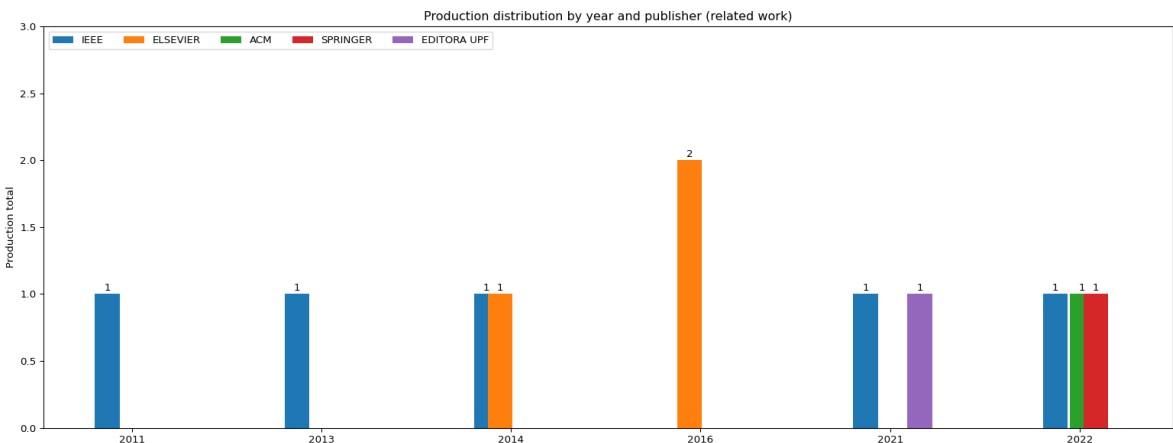

**Figure 2.** Systematic literature reviews by publisher distributed across the years.

After closer inspection of each, four areas of focus were identified for each study: TDD adoption, Quality, Quality/Productivity, and Practice. The results of such a classification are presented in Table 1. Quality/productivity and TDD practices are both topping the charts with four studies each, whereas TDD adoption is represented by only one study. Quality sits in between them, with two studies. It is notable that none of them are closely related to test smells [11], despite TDD being the practice that prioritizes test generation at the beginning of the software development process. Figure 3 depicts the relationship between the publisher and the study focus.

Continuing from the dataset displayed in Table 1, it provides detailed information on study titles, publication years, tagged classifications, and publishers. Following this, we will delve into the categorized systematic literature reviews content.

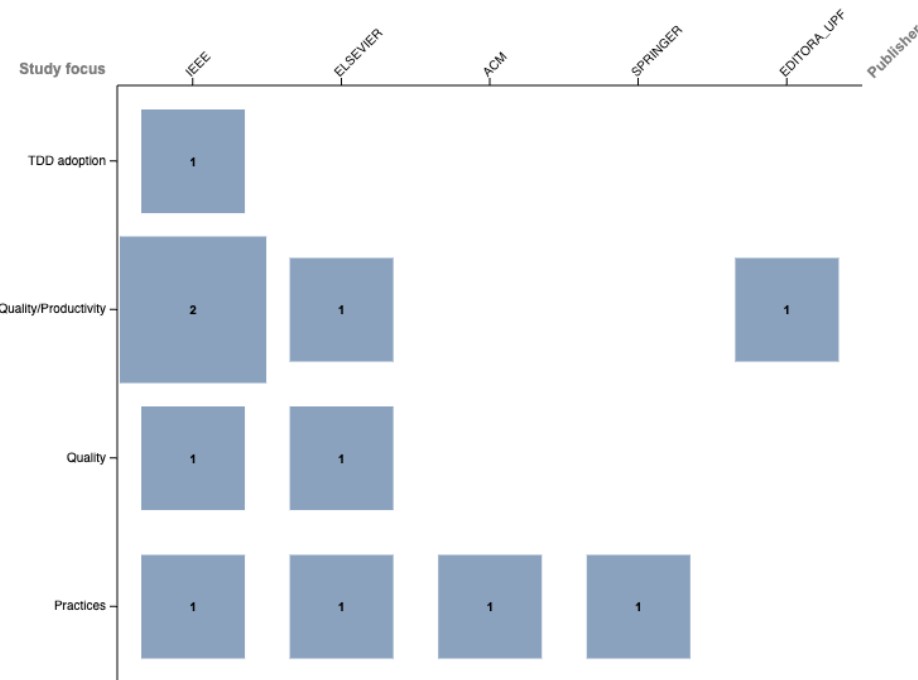

**Figure 3.** Correlation between studies' focus areas and publishers. Chart built with rawgraphs.io.

**Table 1.** Systematic literature reviews in the last 13 years, ordered by year and categorized by study focus area.

| Publication | Focus of the Study | Publication Year | Database | Publisher |
| --- | --- | --- | --- | --- |
| Factors Limiting Industrial Adoption of Test Driven Development—A Systematic [12] Review | TDD adoption | 2011 | Scopus | IEEE |
| The effects of test-driven development on external quality and productivity—A meta-analysis [13] | Quality/Productivity | 2013 | Scopus | IEEE |
| Considering rigor and relevance when evaluating test driven development—A systematic review [14] | Quality | 2014 | Scopus | ELSEVIER |
| Test driven development contribution in universities in producing quality software—A systematic review [15] | Quality | 2014 | Scopus | IEEE |
| The effects of test driven development on internal quality, external quality and productivity—A systematic review | Quality/Productivity | 2016 | Web Of Science | ELSEVIER |
| The impacts of agile and lean practices on project constraints—A tertiary study [16] | Practices | 2016 | Web Of Science | ELSEVIER |
| Test-Driven Development a systematic review [17] | Quality/Productivity | 2020 | Editora UPF [1] | RBCA [2] |
| The effect of Test-Driven Development and Behavior-Driven Development on Project Success Factors—A Systematic Literature Review Based Study [18] | Quality/Productivity | 2021 | Scopus | IEEE |
| Overlap between Automated Unit and Acceptance Testing—A Systematic Literature Review [19] | Practices | 2022 | Scopus | ACM |
| Software Practices For Agile Developers—A Systematic Literature Review [20] | Practices | 2022 | Scopus | IEEE |
| Identifying Guidelines for Test-Driven Development in Software Engineering—A Literature Review [21] | Practices | 2022 | Web Of Science | SPRIGER |

[1] Editora Universidade de Passo Fundo (University of Passo Fundo Press). [2] Revista Brasileira de Computação Aplicada (Brazilian Journal of Applied Computing).

Rafique and Mišić [13] conducted an in-depth investigation of 27 papers. Through their analysis, three categories were identified: context, rigor, and outputs. They found that

TDD results showed a small improvement in quality, but findings regarding productivity were inconclusive. In that context, quality was measured based on the following criteria:

- Number of defects
- Defects per KLOC/defect density
- Percentage of acceptance (external tests) passed
- Percentage of unit tests passed
- Quality mark given by client

The Productivity was measured with:

- Development time/person hours spent/task time
- Lines of code (LOC) divided by effort, LOC per hour
- Total non-commented LOC
- Number of delivered stories per unit effort (or implemented user stories per hour)
- Delivered non-commented LOC per unit development effort (or effort per ideal programming hour)
- Hours per feature/development effort per LOC

The authors also highlight that experienced developers took less time to finish a task in comparison with a novice group. This is also reported by Causevic et al. [12] as a factor that novices have less experience in TDD. Less experience might lead to defaulting back to writing the production code first and testing it afterward, which is also referred to as test last development (TLD).

In 2014, Munir et al. [14] focused on a comparison between TDD and TLD. Test last development (TLD) occurs after the feature has been implemented, meaning that new test cases are added after the programmer has decided that there is enough coverage. This paper shows that with TDD, external quality is positively influenced. In the research context, external quality was categorized into two variables: number of defects and number of test cases. Fewer bugs and more tests mean better external quality.

On the other end of the spectrum, Yahya and Awang Abu Bakar [15] investigated perceptions towards TDD among students. It is known from academia that students often perceive testing activities as boring [22]. However, this paper found that student perceptions are mixed. While some students prefer TDD, others express a preference for TLD. Nevertheless, students indicated their willingness to use TDD if required for joining industry projects.

In 2016, Bissi et al. [16] investigated the effects of TDD across three dimensions: internal quality, external quality, and productivity. The results indicate that TDD is widely practiced in academia (48.14%) and industry (44.46%). The study analyzed a total of 27 papers published between 1999 and 2014, comprising 57.14% experiments and 32.14% case studies. Of these, 76% reported increased internal quality, measured by code coverage—indicating higher coverage implies better internal quality. External quality was also positively impacted, with 88% of the studies reporting improvements, measured through black box testing. However, 44% of the studies reported lower productivity with TDD, as measured by the time taken to implement new functionality.

A tertiary study conducted by Nurdiani et al. [23] focused on agile practices that impact project constraints in software. They found that TDD is the most studied agile practice, and among the studies included in the research, some focus on quality and productivity.

In 2021, Benato and Vilela [17] were interested in the productivity effects promised by TDD and in the quality of the software produced. Through a 17-year analysis of publications (between 2003 and 2020), they found that:

- 54% of studies show an improvement in code quality using TDD—no paper presented a loss of quality.
- 27% presented a decrease in productivity.
- 27% were inconclusive regarding the decrease in quality.

TDD is also a subject that is of interest in the agile community, and it is often combined with different practices, such as behavior-driven development (BDD). In that sense, BDD, like TDD, is composed of stages, but the difference lies in the way those steps are followed. In BDD, the focus is on writing tests in plain text, and it is often not solely composed by developers; rather, it is a team effort.

The benefit of BDD lies in its collaborative aspect, allowing individuals representing the business to write test cases for the application. In their research, Abushama et al. [18] examined various aspects of TDD and BDD, analyzing their differences in project success. Specifically, they investigated development time, cost, and external quality (which, in their context, represents customer satisfaction). The findings are presented in Table 2.

**Table 2.** Results comparing TDD and BDD in terms of: Development Time, Cost and External Quality.

|  | **Development Time** | **Cost** | **External Quality** |
|---|---|---|---|
| TDD | 7 studies showed decrease in time of development, 14 reported increase in the time of development | 4 showed reduced time of debugging | 3 reported positive impact, 5 reported negative impact |
| BDD | negative effect using BDD in comparison with TLD | expensive, one study said it increases the project overall cost | improves external quality in comparison with TLD |

They concluded that TDD and BDD take more time to produce software compared to TLD. Additionally, BDD achieves higher external quality compared to TDD and TLD.

The process of adopting TDD is challenging in both academic settings and industrial ones and the challenges in the industry are different from those faced when adopting it in academia. Causevic et al. [12] enumerated the limiting factors that industrial projects face, and they are as follows:

- Increased development time—The time taken into account was based on the time that it takes to implement a set of requirements, the authors also said that there is a discussion about the time taken for rework of later stages should be included or not
- Insufficient TDD experience/knowledge—Their findings concluded that there were differences between experienced developers and novice developers apply TDD such that they expect that the lack of knowledge in TDD affects its adoption.
- Lack of upfront design—one study reported architectural problems occurring while using TDD, despite TDD being focused on taking small steps and continuously refactoring in order to prevent them.
- Lack of developer skill in writing test cases—out of two studies in this category, one in an industrial context did report the lack of testing knowledge was a limiting factor.
- Insufficient adherence to TDD protocol—five studies with professionals report negative experiences with adhering to TDD protocol. Three of them reported that the reasons include time pressure, lack of discipline, and a shortage of perceived benefits. Two of them reported a correlation between low adherence and low quality.
- Domain and tool-specific issues—out of nine studies, five conducted in an industrial context reported negative experiences. The most commonly reported problem was related to automatically testing graphical user interfaces.
- Legacy code—two studies in an industrial context reported problems while trying to adopt TDD in a legacy codebase. The reason for this was the lack of regression testing, as the codebase does not follow TDD, resulting in missing regression tests

Their work opened a broader discussion over the reasons for resistance in industrial contexts and took a step back from assessing the productivity/quality of the work generated using TDD.

Among the studies related to this work, we also found the work by Staegemann et al. [21], which presented twenty guidelines based on a structured literature review to guide practi-

tioners on the path towards TDD adoption-based on scientific production. The guidelines are cross-functional and cover learning practices that may prevent long-term issues when practicing TDD.

As TDD is a broad subject and touches on different aspects of software development, van Heugten Breurkes et al. [19] found that little research regarding testing practices explicitly investigates both a unit and an acceptance testing perspective. Their work gathered studies from 2013 to 2023. They found that TDD applied to both unit and acceptance testing encourages developers to specifically consider design, including software features, and to complete work in small increments. Additionally, TDD can be applied to broader scopes of domains, including scientific computing and embedded systems. Lastly, the literature warned about the learning curve (e.g., refactoring, resistance to change in mindset).

Often, TDD is a practice that is used in combination with other aspects of software engineering, such as domain driven design (DDD), model-driven development (MDD), and BDD. DDD and MDD focus on software modeling, looking at it from the business perspective first [20]. For TDD practitioners, such an approach brings benefits, given that understanding the business is a core activity to meet requirements definition.

Despite focusing on different aspects of TDD (as previously mentioned: adoption, quality, productivity, and practice), the presented research explores TDD itself, its outcomes, and how to adopt it within certain guidelines to improve the practice's success. Based on the 13 years of related literature analyzed in this section, there is a gap in combining the practice of TDD in teams involves the generation of test smells/anti-patterns in the codebase. To that end, this study presents the process and results of a systematic literature review [24] focused on TDD and the common factors that lead to TDD anti-patterns/test smells in an industrial context.

Transitioning from the insights gleaned in the related work, the next section focuses on the methodology employed in this research.

## 3. Methodology

In this section, we delve into the methodology employed to conduct the systematic literature review, drawing upon the PRISMA framework [25]. We begin by outlining the protocol utilized, followed by the process for filtering studies. Finally, we summarize the application of PRISMA and discuss the challenges encountered in extracting and formatting the dataset.

Our starting point was the formulation of the following research questions:

- RQ1: When developers practice TDD, are there external influences on developers to use TDD?
- RQ2: Does the practice of TDD influence the addition or maintenance of new test cases?

In the next subsection, we explore the breakdown of those research questions into the protocol itself.

### 3.1. The Protocol

3.1.1. Identifying Primary Studies

In this phase, the identification of primary studies utilized the PICOC (population, intervention, comparison, outcome, and context) framework in an attempt to implement evidence-based practice. Each letter in PICOC stands for [26]:

- Population: Who?
- Intervention: What or How?
- Comparison: Compared to what?
- Outcome: What are you trying to accomplish/improve?
- Context: In what kind of organization/circumstances?

Taking into account the PICOC definition, the presented study focuses on professionals practicing TDD. In that sense, the PICOC generated for this research is presented in Table 3.

**Table 3.** PICOC definition with the research context in place.

| Population | Intervention | Comparison | Outcome | Context |
|---|---|---|---|---|
| developers | TDD | | anti-patterns | Preferably in professional settings |
| practitioners | Test-Driven Development | | pain points | |
| | Test-Driven Development unit test unit testing | | test smells | |

The context of the research currently prevents any kind of comparison. Furthermore, the objective is not to compare any existing practices with TDD; rather, the focus is to understand what leads to test smells/anti-patterns.

### 3.1.2. Data Source

Based on the PICOC definition, we began searching for databases to retrieve the studies. Ultimately, we concluded on three of them: Scopus, Web of Science, and Google Scholar. The rationale behind these choices is as follows:

- Google Scholar was selected due to its association with Google, as it is the search engine commonly used by practitioners. Additionally, its indexing system includes publications not found in scientific databases. The selection of this database was based on the PICOC criteria. Despite the lack of transparency regarding the inner workings of the search engine [27], Google Scholar is recommended as a source to be used in conjunction with other databases for systematic literature. reviews due to its extensive reach for studies in the gray literature [28]
- Web of Science and Scopus were selected because both aggregate scientific work across various types of publications.

Using those databases combines the gray literature with the scientific work produced in the targeted research area. To extract information from these databases, three different search strings were built based on the PICOC, taking into account the following criteria:

- TDD—the keyword was added as is.
- Test-driven development—the keyword was added as is, and the variation with a hyphen was also included.
- unit test—it was included given that there are studies that use unit tests instead of TDD, even though they are closely related.
- pain points—the keyword was not used, as it does not refer to the subject of interest in this research. The keywords anti-patterns and test smells were used instead.
- anti-patterns—the keyword was included as is, and its variations were also taken into account.
- test smells—the keyword was included as is, and its variations were also taken into account.
- developers/practitioners—the keyword was not added as it is not the subject of the study. This point was taken into account in Section 3.2.4.

Combining both the criteria and the PICOC resulted in the search strings used in each database. Each string was tested to find a balance between the number of results and the studies that best fit the criteria, resulting in the following search strings for Scopus:

TITLE-ABS-KEY ( ( tdd OR "unit test*" OR "test-driven development" OR "test driven development" ) AND ( "anti-pattern*" OR "anti pattern*" OR antipattern* OR pattern* OR "test smells" ) )

For Web of Science:

> TS=( ( tdd OR "unit test*" OR "test-driven development" OR "test driven development" ) AND ( "anti-pattern*" OR "anti pattern*" OR antipattern* OR pattern* OR "test smell*" ) )

For Google Scholar:

> (tdd OR "unit test*" OR "test-driven development" OR "test driven development" ) AND ( "anti-pattern*" OR "anti pattern*" OR antipattern* OR pattern* )

Table 4 displays the number of papers found in each database based on each search string. The numbers shown were gathered in October 2023:

**Table 4.** Number of published studies by database.

| Database | Number of Papers |
|---|---|
| Scopus | 780 |
| Web of Science | 302 |
| Google Scholar | 954 |
| Total | 2036 |

As the first step of analyzing the dataset, the software Bibliometrix [29] and VosViewer [30] were used to plot the information for Web Of Science and Scopus. Google Scholar was excluded, as both software do not support it. Figure 4 illustrates the most relevant affiliations, Figure 5 shows the most relevant sources, Figure 6 displays the most relevant authors for this field of study, and Figure 7 represents the citation network.

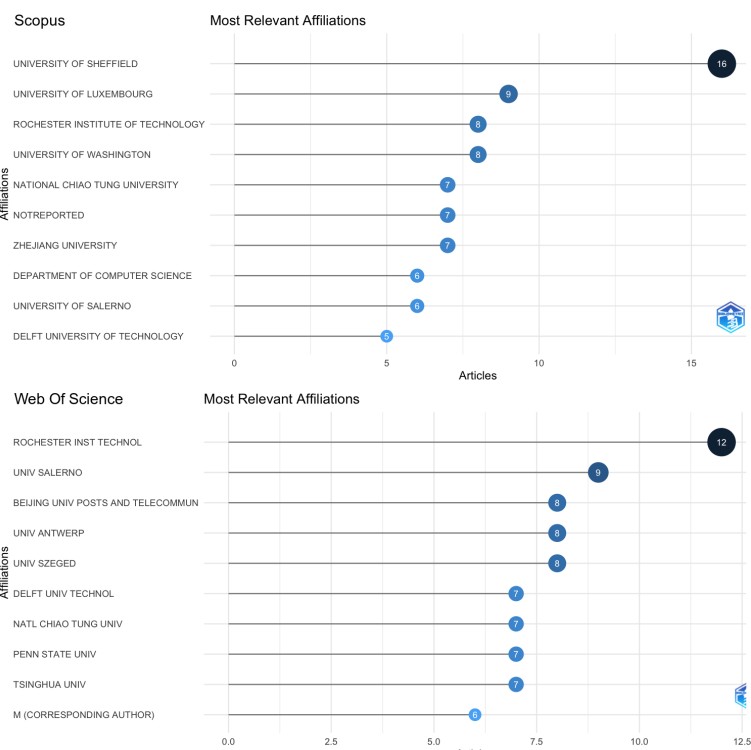

**Figure 4.** Most relevant affiliations in Scopus and Web Of Science.

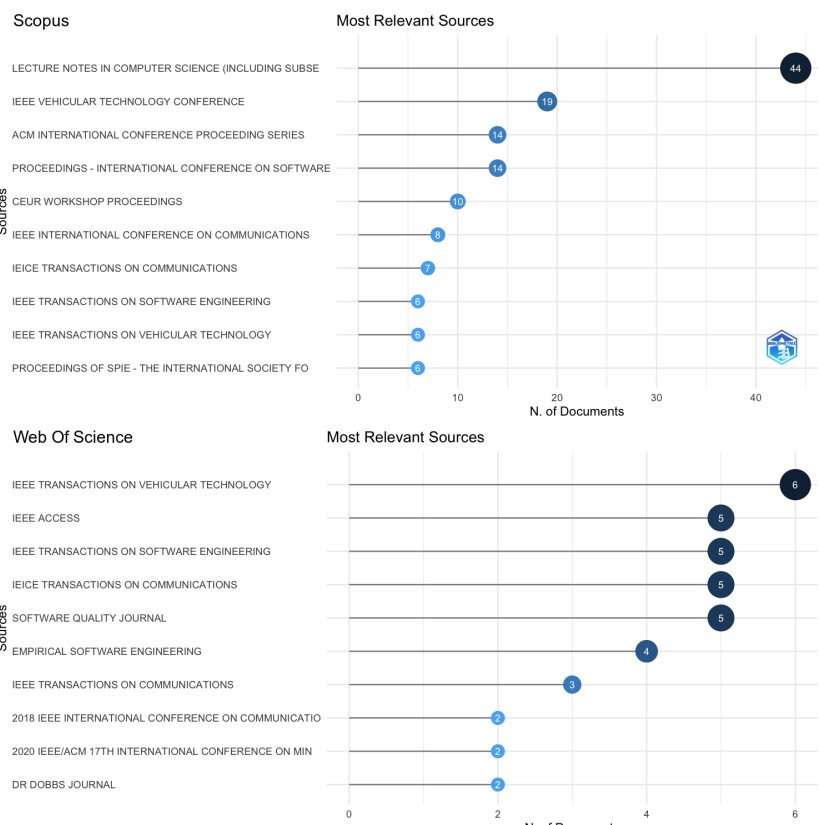

**Figure 5.** Most relevant sources in Scopus and Web Of Science.

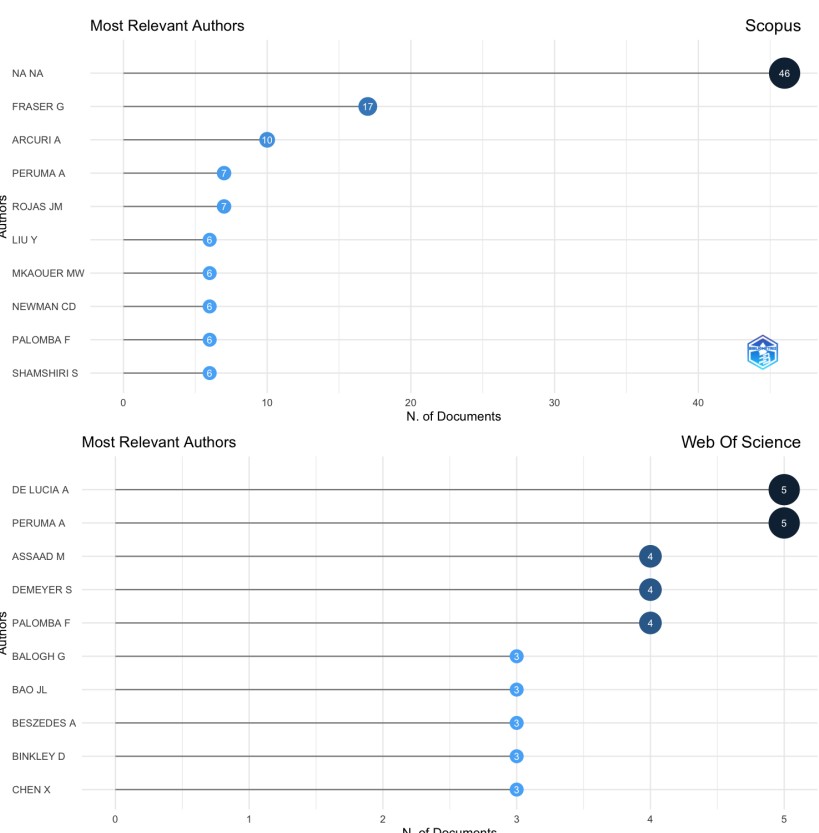

**Figure 6.** Most relevant authors in Scopus and Web Of Science.

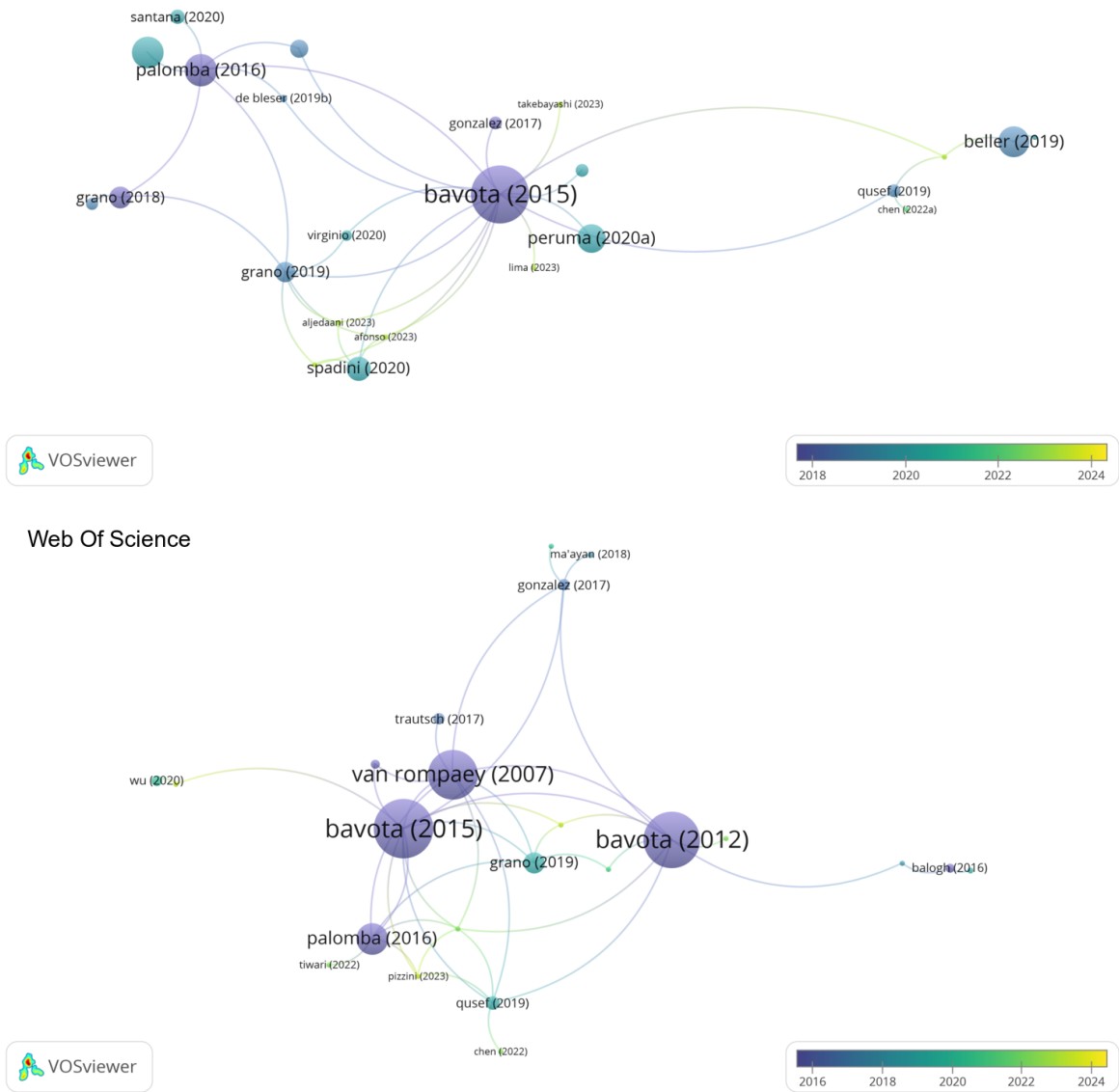

**Figure 7.** Citation network in Scopus and Web Of Science.

In order to plot those charts, filters were added to exclude areas that are not related to computer science or software engineering. In Scopus and Web of Science, TDD is an acronym used in other areas of knowledge. For example, records have been found in physics, chemical engineering, and neuroscience, just to name a few. Despite having textual matches, in each area of knowledge, it means different things. Thus, the decision was made to remove those records before plotting the charts. The strings and dataset used to plot the charts are available in the GitHub repository: https://github.com/marabesi/slr-tdd-anti-patterns-test-smells (accessed on 14 March 2024).

*3.2. Review Process—Filtering*

3.2.1. Inclusion

- CI1—Papers that mention TDD as a software development practice
- CI2—Papers that have been published in Portuguese, Spanish, or English; these are the known languages of the authors
- CI3—Papers published in conferences, journals, technical reports, and not proceedings

- CI4—Papers published in the gray literature
- CI5—Most current paper (if duplicates)
- CI6—Papers that are available for access
- CI7—Papers that are related to the area of computing (and not biochemistry and others)

### 3.2.2. Exclusion

- CE1—Papers that do not mention TDD as a practice
- CE2—Papers that are not in Portuguese, Spanish, or English
- CE3—Papers that have not been published in conferences, journals, technical reports, or proceedings
- CE4—Papers that have not been published in the defined gray sources
- CE5—Old versions of duplicate papers
- CE6—Papers that cannot be accessed
- CE7—Papers that are not within the computing theme (papers that are in the medicine database, for example)

### 3.2.3. Study Selection Criteria

At this stage, duplicates were removed according to CE5 criteria. To identify duplicates, this study utilized a two-phase analysis. The first phase employed a Google Docs extension called Remove Duplicates (listed in the Google Workspace Marketplace). This extension provides mechanisms for identifying duplicates based on spreadsheet columns. When a duplicate is detected, it is tagged in a new column.

The second phase involved an automated duplicate analysis provided by rayyan.ai https://www.rayyan.ai (accessed on 10 March 2024). This phase was added because manual inspection revealed that some duplicates were not properly tagged. This was due to the extension's limitation of using only the title of the study as a criterion for duplicate detection. Rayyan offers an automated detection mechanism that flags the likelihood of a study being a duplicate based on title, author, journal, and year. The tool's help section assists users in navigating the detection feature: https://help.rayyan.ai/hc/en-us/articles/4408790828689-Detecting-duplicate-references-articles-in-a-review (accessed on 10 March 2024).

After both approaches were completed, the lists of duplicates from each phase were merged, and manual inspection commenced to verify the results, whenever a duplicate was identified, the study with the fewer records in the database was retained. Table 5 displays the number of duplicated studies removed after this process.

**Table 5.** Number of unique studies vs duplicated studies by database.

| Database | Accepted | Duplicated |
|---|---|---|
| Scopus | 494 | 286 |
| Web of Science | 288 | 14 |
| Google Scholar | 909 | 45 |
| Total | 1691 | 345 |

Once the papers were cleaned up, removing the duplicates, a process of reviewing each paper and applying the inclusion and exclusion criteria started. Table 6 depicts the resulting numbers after applying the criteria. Note that the columns CE4 and CE5 have no records. CE4 is due to the classification that could not find any matching document, while CE5 is due to the removal of duplicates as defined in the previous step.

Once the criteria were applied, an initial analysis of the production started. A total of 272 studies were included to undergo a more detailed analysis. The distribution of studies over the years shows that the work produced was more active in the last ten years. A detailed view is provided by Figure 8.

**Table 6.** Number of studies categorized by Inclusion criteria and Exclusion criteria by database.

| Database | CI1 | CE1 | CE2 | CE3 | CE4 | CE5 | CE6 | CE7 |
|---|---|---|---|---|---|---|---|---|
| Scopus | 44 | 281 | 0 | 11 | 0 | 0 | 0 | 158 |
| Web of Science | 16 | 91 | 0 | 0 | 0 | 0 | 1 | 180 |
| Google Scholar | 212 | 630 | 37 | 0 | 0 | 0 | 27 | 3 |

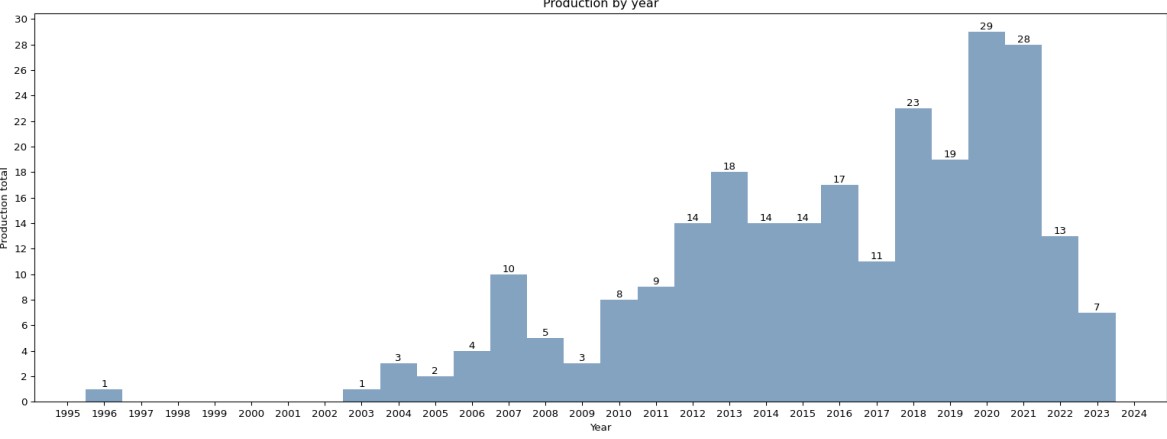

**Figure 8.** Number of studies selected for further analysis published by year.

Given the structure of the gray literature tackled in this study, Google Scholar had the most studies that were accepted, as depicted by Figure 9.

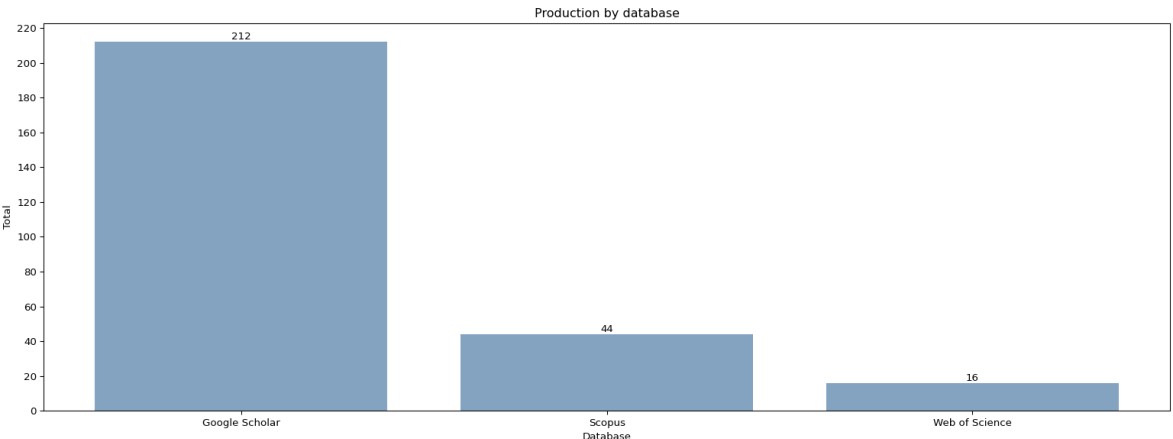

**Figure 9.** Number of studies selected for further analysis by database.

The following section presents the process of mapping the research questions to the selected studies.

### 3.2.4. Criteria for Quality Evaluation

The next step in the research follow-up was to map the research questions formulated for this study and narrow down the number of papers to focus on. We formulated six questions linked to the research proposal:

- Q1—Does the paper presented mention TDD anti-patterns/test smells?
- Q2—Did the present studies do experiments in professional(industry) settings?
- Q3—Do the studies talk about external factors and not just the code (such as the team and how they usually create test cases)?
- Q4—Do studies link team experience to anti-patterns/test smells?
- Q5—Is there a differentiation between TDD inside-out and TDD outside-in?

- Q6—Is there mention of the time that the team practices TDD?

For each question, a weight of 0, 0.5, and 1 was assigned (0 for not scoring any points for the question, 0.5 for partially scoring, and 1 if it fully answered the question). This means that a paper could score a maximum of 6 if it answered all the questions. A higher score indicates that the study likely answered more questions, thus adhering to the research goal, while a lower score means the opposite. Table 7 presents the final score.

**Table 7.** Number of papers based on the scoring.

| Score | Number of Papers |
|-------|------------------|
| 0 | 110 |
| 0.5 | 43 |
| 1 | 54 |
| 1.5 | 37 |
| 2 | 18 |
| 2.5 | 4 |
| 3 | 5 |
| 4 | 1 |
| Total | 272 |

Upon exploring the results of the classification, it was observed that, based on the scoring system, the majority of the studies scored 0, 0.5, and 1. This suggests that, while most studies in this range are relevant to the research topic, they lack comprehensive answers for quality evaluation.

On the opposite end of the spectrum, a handful of studies scored the highest, totaling six studies between 3 and 4. Upon a cursory examination of these six studies, it became apparent that utilizing such a restrictive scoring system would exclude other relevant studies that contribute insights to the research conducted.

As a result, the decision was made to include studies that scored 2 or higher in the quality evaluation, resulting in a dataset of 28 studies.

### 3.3. Resume and Data Extraction

The previous sections depicted the protocol and process that were used to filter the dataset. The same information is summarized in Figure 10.

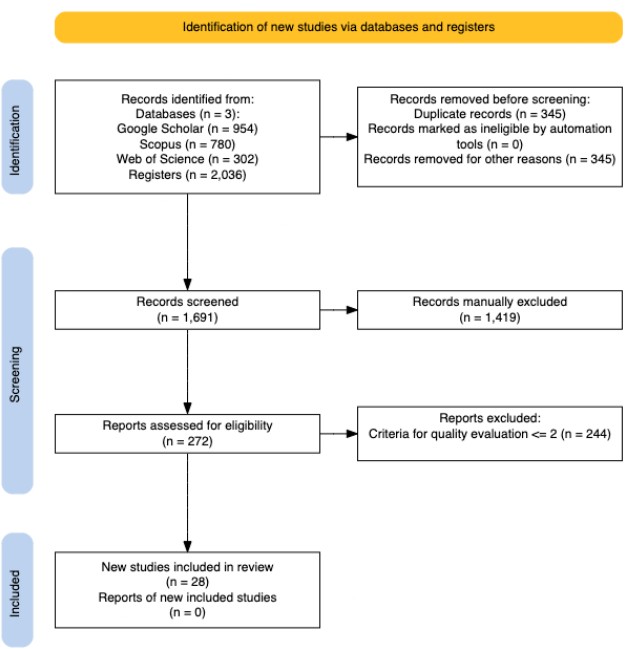

**Figure 10.** PRISMA flow chart based on PRISMA2020 [31].

### 3.3.1. Extracting Information

The study is based on three different databases, which pose challenges for extracting and combining the information required for a systematic literature review. Both Scopus and Web of Science offer functionality to extract the results of a given search in different formats. However, Google Scholar requires an additional step to gather the information, as it does not have such functionality.

To ensure the traceability and reproducibility of the findings, a repository on GitHub https://github.com/marabesi/slr-tdd-anti-patterns-test-smells (accessed on 14 March 2024) was created, and the raw search results are stored in the "databases" folder. For Scopus and Web of Science, the results were downloaded in CSV format and stored in the GitHub repository. However, for Google Scholar, an additional step was required: extracting the results using an open-source library called scholarly [32]. This library utilizes a web crawler to extract information from Google Scholar, providing an API to store the results in JSON format for later use. This approach anticipates issues such as pagination of large numbers of records and blocked IP addresses [28]. To ensure verifiability, the results were stored in the "raw-result-from-google-scholar" folder.

With this information in place, a script named *scholar-search.py* was created to extract information from Google Scholar. The script is available on GitHub: https://github.com/marabesi/slr-tdd-anti-patterns-test-smells/blob/main/scholar-search.py (accessed on 14 March 2024). The script executes the search with the search string, and for each result, it stores the raw JSON for each record it finds. The result was 954 JSON files stored on the disk (and available on GitHub). At this point, having the JSON files enabled the step of CSV conversion to make it compatible with the other two databases. Another script named *parse_to_csv.py* was created to parse the information gathered. The script is available on GitHub: https://github.com/marabesi/slr-tdd-anti-patterns-test-smells/blob/main/parse_to_csv.py (accessed on 14 March 2024). The end result was a CSV file for each database.

### 3.3.2. Clean up Data from Google Scholar

Looking at the results in Google Scholar, it was noticed that certain records had their values changed, making the parser and the reading unreliable. After analyzing the detailed information in the CSV file, it was verified that the field was not symmetrical; that is, some lines had information, and others did not. This asymmetry became a challenge to read and analyze the CSV in two ways:

- The process of reading the CSV by a person is challenging because, as you read it, you have to correct it manually, one by one, and mark if the data is missing.
- Attempting to read the file and inspect its content with the csvkit library https://csvkit.readthedocs.io/en/latest (accessed on 14 March 2024) results in an error when trying to view the fields of a non-symmetric CSV. This issue is specifically related to csvkit; other tools such as Excel can read and parse CSV files without errors.

The solution found was to create a base structure before converting the data from Google Scholar to CSV. The basic structure assumes that no information will be available and tags it as "KO". The resulting dataset is available for public inspection (Dataset with the resulting process of gathering studies from Google Scholar, Scopus, and Web of Science: https://bit.ly/3Ta7p1z (accessed on 14 March 2024).

## 4. Results

The studies selected in this phase cover a wide range of subjects. Initially, each study was analyzed, and categories were extracted based on the focus of the study. For example, if the study was focused on teaching TDD, it would be tagged as TDD/Teaching; if the study was focused on organizational aspects, it was tagged as Broader context, and so on. As the analysis progressed, all the studies were tagged accordingly. In total, five categories were extracted from the studies analyzed. Figure 11 depicts the number of studies for each category.

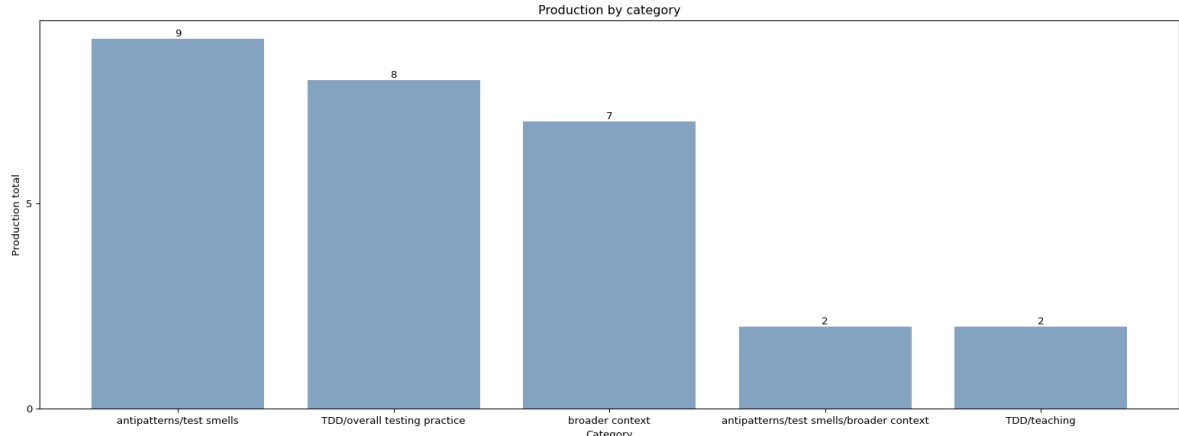

**Figure 11.** Number of studies by categories.

Each category was associated with the mapping questions presented in Section 3.2.4. The Table 8 illustrates the relationship between each category and the corresponding question. The complete list of studies linked to each category is available in the Appendix A.

**Table 8.** Categories based on the quality criteria.

| Category | Associated Question |
| --- | --- |
| Anti-patterns/test smells (Table A3) | Q1—Does the paper presented mention TDD anti-patterns? |
| TDD/overall testing practice (Table A4) | Q6—Is there mention of the time that the team practices TDD? |
| Broader context + TDD/teaching + Anti-patterns/test smells/broader context (Tables A1–A3) | Q2—Did the present studies do experiments in professional(industry) settings? |
| | Q3—Do the studies talk about external factors and not just the code (such as the team and how they usually create test cases)? |
| | Q4—Do studies link team experience to anti-patterns? |
| | Q5—Is there a differentiation between TDD inside-out and TDD outside-in? |

Note that the grouping reduced the original 5 categories to 3. This reduction occurred because the category Broader context serves as an overarching umbrella that encompasses the other two categories. This decision was made after synthesizing the data, enabling a more concise presentation of the results. The challenge encountered in this step was to present the data in a comprehensible manner. The following list outlines the rationale behind grouping the categories into three:

- Broader context + TDD/teaching + Anti-patterns/test smells—In this category, other areas related to the focus of the study were added. For example, testing strategies and the DevOps maturity model were included.
- TDD/overall testing practice—Studies in this section refer to the practice of TDD that is not directly related to test smells. In this category, studies that mentioned TDD as a practice in any form were added.
- Anti-patterns/test smells—This category includes studies that mention anti-patterns or test smells in the study. Most of the studies specifically mention test smells rather than anti-patterns. For the purposes of this section, both were considered.

The following sections provide detailed descriptions of the results for each category defined above.

## 4.1. Broader Context, TDD/Teaching and Anti-Patterns/Test Smells

Test smells are a well-researched subject that is accompanied by various perspectives and testing strategies. Researchers have made efforts to discover the effects of test smells in learning and industrial environments, as well as their impacts on code maintainability

and understandability. Therefore, before delving into the test smells themselves, this first section focuses on the findings related to the context in which students and professionals develop software. It encompasses different types of applications, from software deployed on client premises to web applications.

### 4.1.1. Software Development

A survey involving 456 participants was conducted across the industry with the goal of gaining insight into how they align with industry-standard processes. To achieve this, the survey was shared with individuals from IT companies and meetup groups and posted on public forums [33].

The respondents covered a range of roles, including developers, executives, technical writers, testers, line managers, architects, team leads, business operations professionals, and project managers. Consequently, the analysis of the results was categorized based on three factors: role, company size, and experience level. The following list summarizes the results by topics of interest:

- Anti-patterns: In this survey, "anti-patterns" have a different meaning and are not specifically related to testing anti-patterns or test smells.
    - Most respondents have never heard of or are not concerned about anti-patterns.
    - Developers understand anti-patterns the most, while testers understand them the least.
- Gaining knowledge (The top three sources by respondents)
    - Internet forums and blogs (82%)
    - Colleagues (79%)
    - Books (65%)
- Static analysis and traceability
    - Manual code reviews are used in internally developed products by: 81% of developers, 67% of testers, 74% of managers, and 39% of technical writers.
    - The average time of manual code reviews was more than 50 min across roles.

Regarding results based on the size of the companies, the study found that larger companies utilize their resources more efficiently without indirectly forcing people to produce lower-quality results. Regarding testing specifically, results across experience levels suggest that knowledge of design patterns, testing, and management techniques does not depend on the respondents' level of work experience.

The findings regarding code review and the connection with the time taken to manually approve changes might indicate a lack of trying other ways to integrate work in an agile context. Automation of such tasks should be central to the process, where feasible. In contexts where possible, pair programming and trunk-based development [34] serve as an alternative. Research suggests that by combining pair programming with other practices, achieving a sustainable pace for software development is possible [35].

The automation aspect of the testing practices to support an agile environment is also a subject of discussion in academia. According to Hartikainen [36] automation is one of the key concepts in agile testing. Therefore, the practice of testing varies based on the organization that applies it. In their study, the 5 levels of maturity that are listed as follows are used to define different levels of testing. Each level is defined by three categories of testing: unit, integration, and system.

- Base—only units are automated
- Beginner—unit fully automated and integration testing is partially automated
- Intermediate—unit, integration is fully automated and the system is partially automated
- Advanced—unit, integration is fully automated and the system is partially automated. At this level, non-functional testing is supported by system testing.

- Extreme—unit, integration is fully automated and the system is partially automated. At this level, non-functional testing is supported by system testing and integration testing.

Note that in the list, from top to bottom, the changes that are applied refer to the number of steps automated. For an agile context, it is desirable to have as much automation as possible. In their research, the maturity model is taken a step further, and the context of a web application is used to depict what factors are taken into account to define a test strategy. The results suggest four factors that are used to define a proper strategy: Development and management team, Requirements, Customer, and Technology. Each one of them has subfactors associated, as follows:

- Development and management team
  - Known-how—Experience, Testing tool and practice, Knowledge, Testability of the code, Business domain
  - Size
  - Understanding the value of testing
  - Testing interest and motivation
- Customer
  - Understanding of software development
  - Understanding the value of testing
  - Willingness
- Technology
  - Tooling—Availability, Quality
  - Development technology—Limitations
- Requirements
  - Complexity—Use cases, Business domain, User interface, Integrations to external systems, Access levels, Build and test process
  - Changeability
  - Non-functional
  - Defect liability
  - Specificity
  - Application—UI or server oriented, Type of end user(business/consumer), Lifespan, New/Legacy, Size, Expected number of users, Expected number of deployments
  - Project requirements—Budget, Schedule, Iterativeness
  - Criticality—Application, Business

The authors suggested that the best testing decisions for this study case are those considered within the organization, project, and application context mentioned. Combining the maturity model and the factors found is an option to help make decisions for a testing strategy.

Automation and testing are at the core of the DevOps movement. One of the factors that made DevOps a popular subject was the book "The Phoenix Project" [37], where the notion of unifying the organization's goals with IT goals were already in place. Researchers also tackled the challenge of understanding where organizations stand regarding DevOps. For instance, Kurkela [38] conducted a study to determine the state of DevOps within a team of 17 members. The team was split into two different geographical locations and worked to deliver software installed on customers' premises.

The initial analysis of the team's ways of working shows that the team uses SCRUM as a framework to develop software, but the framework is not utilized to its full capacity. The framework itself does not have any formal recommendations regarding testing practices. Therefore, in the literature, Tarlinder [39] presented a perspective to fill in this gap. The author utilizes the maturity model developed by Zarour et al. [40], which consists of five levels of maturity. Once again, automation is one of the key aspects of a higher maturity model, as well as found by Hartikainen [36]. However, the difference lies in the fact that

automation is targeted at the organizational level rather than solely for testing purposes. The main findings regarding the model are as follows:

- Quality—Quality standards are not shared between other teams.
- Automation—Automation does not have organization level guidance; the team still uses a continuous integration pipeline.
- Communication and collaboration—The team follows a hierarchical model.
- Governance—The development process is not followed as documented.

As a final remark, the author also suggested that the team should shift-left the testing process as an opportunity to create better test cases. However, at the time of publication, the requirements were not always clear enough to support testing. Requirements have their own research branch that researchers focus on. Furthermore, they are at the core of an effective strategy for software testing. It is reported that testing activities for software consume 75% of the software development process [41].

Test efforts are a subject that interests decision-makers in the software development process, which, in turn, makes the predictability of efforts a matter of interest. Torkar et al. [42] presented a model for project managers to use when predicting software test effort in iterative development.

The tool features a Bayesian network and combines different data types to develop predictions. The researchers used and validated the tool in two industry projects. The results indicate a model that contributes to accurate predictions based on data fed into the model for both projects. The authors recommended the tool to assist test managers in planning and controlling test activities.

4.1.2. Learning TDD

Teaching TDD in a non-professional context requires simulated problems to help students grasp the basics of TDD. Parsons et al. [43] state that it is important to gain an understanding of TDD through the experiences of those who have successfully adopted this practice, given the complexity of the subject for industrial projects. In that regard, students might also benefit from those experiences to enhance their learning experience.

Considering this scenario, Dimitrichka [41] addresses various aspects of teaching automated testing. In their work, it is stated that between 50% and 75% of a company's time is spent on testing activities, and not only that, 40% of a company's budget is spent as well. This builds the argument towards a more automated approach for testing, leading to the need for training in this field. Besides, automation is used to measure different maturity models [36,40].

The testing activity has many facets: business requirements, assessing usability, assessing performance, assessing security, and assessing compatibility. Depending on which facet is targeted, different levels of automation and techniques are required.

Unit testing fits within the business requirements facet. Taking into account the test pyramid [44], unit testing should be the initial step to automate a software project. This is often the reason to incorporate test-driven development (TDD) as a practice from the start in the software development process.

The training of TDD becomes a key player, as a lack of it might lead to anti-patterns. In their work, Dimitrichka [41] identified four anti-patterns resulting from inexperience or a lack of knowledge in resolving certain types of problems. They connect the following anti-patterns:

- Loudmouth—Polluting the test output leads to questioning if the test passed for the right reason.
- Greedy Catcher—Catching exceptions just to make a test pass.
- Sequencer—Avoid coupling test cases with the order in which they appear in a list.
- Enumerator—Naming test cases are used as a way of debugging and quickly spotting problems; naming them randomly harms understandability.

- The Liar—A test that does not fail. It is often related with async-oriented or time-oriented to prevent false positives.

From the list, four (Loudmouth, Sequencer, Enumerator, The Liar) out of five fit the first level of anti-patterns identified by Marabesi et al. [45]. The liar, according to Dimitrichka [41], is one of the most harmful TDD anti-patterns. By definition, it does not test what it is supposed to test, and as a consequence, the test does not fail. Note that this study uses anti-patterns rather than test smells to refer to the code, whereas Bai et al. [46] investigated students' perceptions about unit testing and used test smells to identify the most common ones. In their study, they found that students value the following:

- Code coverage is the aspect most important for students.
- Understanding the reason behind flaky tests and fixing them is challenging.
- Students struggle to find what makes a unit test good.
- Six test smells are also found to be the most common: Test Redundancy, Bad Naming, Lack Comments, Happy Path Only, Refused Bequest, and No Assertions.

If teaching TDD presents its own challenges for academics to address in an isolated environment, the failure to deliver well-thought-out training can have repercussions in various aspects of students' professional careers. On one hand, students might carry their gaps into the industry and address them there, or in the worst-case scenario, these gaps may persist, leading to test suites that do not provide the best feedback for maintenance activities.

An alternative for professionals who have not had the chance to undergo proper training in TDD is to learn it in their daily work at their companies. In this scenario, the challenge is twofold, as the context is not solely for learning purposes; it is also implicit that it should drive business outcomes. In a report made by Law [47] on the lessons learned from adopting TDD in two teams at an energy company, there is a different context and learning experience compared to those faced in the classroom with students.

Two teams were scheduled to try adopting TDD, with the project's goal being the migration of a legacy system to a new Java-based version. In one team, there was no previous experience in adopting TDD, so the required skills were built through self-learning media (such as forums, books, and vendor sites, reported by Kovács [33] as one of the primary methods of gaining new knowledge). On the other team, practitioners already had exposure to the TDD practice. This experience proved valuable in sharing practices with others as well as laying a foundation for the project. For this case study, the success of the test-driven approach was a combination of different factors:

- Experienced team champion
  - It enables knowledge sharing across individuals and teams.
- Well-defined test scope
  - Different types of tests are recommended, such as unit, integration, and acceptance tests. Deciding which one to pick is a discussion among developers who are implementing the functionality. The question of when to stop or continue should be based on business value.
- Supportive database environment
  - For this specific case, a shared database was used to avoid complexities in the environment setup and to generate many database instances just for the sake of testing; the participants took advantage of the fact that they were on site. This is another advantage that relates to eXtreme Programmimg practices, as the team is colocated and such synchronization becomes instant.
- Repeatable software design pattern
  - Designing, building, and maintaining support for data generation tests was one of the challenges reported by participants. A design pattern called "Object Mother" [48] was mentioned for reusing test objects.

- Complementary manual testing—Despite how much testing is performed, there is no guarantee that an application is free of defects. As such, automated testing is a priority. However, regardless of how extensively automated tests are developed, manual tests must still be performed.

### 4.1.3. TDD Practice

Whether tackled from an academic perspective or in professional settings, TDD is being applied in various contexts and in combination with other agile practices [43]. This leads practitioners to advocate that TDD leads to better design, as the practice prioritizes testability first.

In a study by [4], the influence that TDD has on the creation of classes in Object-Oriented Programming (OOP) based projects were explored. The approach taken was twofold: first, a qualitative exploratory study was conducted, with 25 participants across 6 different companies implementing exercises using TDD. Java was used, and the minimum requirement was to have knowledge of writing tests. As a follow-up, experts in the field were invited to categorize the code into three categories: simplicity, testability, and quality of class project. None of them knew which code was written with TDD or not.

On one hand, the results suggest that regarding quantitative code metrics such as code complexity, fan-out, cohesion, and the number of lines per method, there is no difference with or without TDD. On the other hand, the qualitative results suggest that, even though the practice of TDD does not directly lead to better class design, participants feel that the practice of TDD brings benefits, and it is difficult to stop once started. This points to the following:

- Using TDD provides constant feedback.
- TDD enables testability, and if testing is difficult, it indicates a code smell that needs refactoring.

Regardless of the empirical evidence presented for class design, TDD is practiced by practitioners on a daily basis, and there are other aspects of the practice that researchers have investigated. A common trait among practitioners as they gain experience in writing tests is to ask the question, 'How good are my tests?' This reflective question aims at improving one's understanding of the subject.

Bowes et al. [49] researched the topic of characterizing what makes a good test through a two-day workshop conducted with industrial partners to investigate test principles. They combined this with relevant content from practitioners' books and surveyed existing literature. Kent Beck also shared his thoughts on what makes test more valuable: https://kentbeck.github.io/TestDesiderata (accessed on 1 February 2024). As a result of this experiment, the authors created a list of 15 principles enumerated as follows and related them to test smells.

- Simplicity
- Readability and Comprehension
- Single Responsibility
- Avoid Over-protectiveness
- Test behaviour (not implementation)
- Maintainability: refactoring test code
- A test should fail
- Reliability
- Happy vs Sad test
- Test should not dictate code
- Fast feedback
- 4 phase test design
- Simplicity of fixtures
- Test independency/ Test dependency
- Use of test doubles

The main contribution of having these principles, according to the authors, is to look at different aspects of quality, not only focusing on coverage and effectiveness as previously depicted in the study. Interestingly enough, coverage is one of the aspects that students value the most [46].

Practitioners might face different challenges when incorporating test-driven development into software development process. One of the most challenging aspects might be incorporating the practice into an application that was not originally written with this practice in mind. The work by Wall [50] reports a case study on the testability aspects of adding tests to an existing application and its implications. One of the first observations made by the author is that adding tests to code that was written without testing in mind is challenging. This means writing the tests based on details about the implementation rather than basing them on what the system should do.

TDD can be applied even to an application that previously had no tests. Therefore, writing tests for an application without tests requires changing the application to meet testability criteria, and many changes are most likely to have an impact on the design, leading to a refactoring to accommodate the change [51].

When deciding what to test, it pays off to focus on parts that are central to how the application is used; the most critical aspects should be tackled first. An option to find where this spot lies is to use metrics derived from the version control system [52].

*4.2. Anti-Patterns/Test Smells*

There is a consensus in the literature that Van Deursen et al. [11] listed the set of test smells used as a base for different studies [7,49,53–57]. Still, there seems to be a discrepancy between academia and the gray literature, as different sources that professional software developers consume point in another direction. Even the mix of test smells and anti-patterns seems to be in a gray area. The work done by Garousi and Küçük [58] combined the available sources and definitions of test smells in a single place.

Test smells were also the subject of research on manual test case creation. The work by Hauptmann et al. [54] focused on tests written in natural language. Tests written in natural language often do not adhere to well-established software engineering principles due to their manual creation, resulting in each test suite likely being different from one another. This discrepancy leads to defects affecting maintenance and comprehension. As a possible solution to this issue, they proposed metrics to measure the quality of natural language tests.

Those metrics are closely related to the clean code principles outlined in the book by Robert [59], such as hard-coded values, long test step names, and inconsistent wording. It is important to note that there are a total of 7 metrics that the authors used to measure the test suites. To put the metrics to the test, seven projects from two companies were selected for study. With this sample of applications, the analyzed test suites provide substantially diverse functionality, ranging from damage prediction and pharmaceutical risk management to credit and company structure administration.

The results of the experiment revealed that each test suite exhibits test smells, although the range of detectable smells varies between them. In addition to identifying these test smells, the authors highlight an insight: the results alone cannot provide a definitive assessment of test suite quality. Other factors should also be considered. Nevertheless, these metrics serve as an initial step toward continuously improving the quality of the test suite.

### 4.2.1. Test Smells across the Industry

Despite the body of knowledge surrounding test smells, there is a lack of awareness of them in the industry. The research conducted by Kummer et al. [60] aimed to assess the level of familiarity developers have with test smells, as well as the frequency and severity of these smells in an industrial context. Both qualitative and quantitative methods were employed to gather data through a survey. As a follow-up, interviews were conducted with

20 individuals with varying levels of experience in the industry. The results indicate that while participants were sometimes able to describe a test smell mentioned in the literature, they were unable to provide its name.

In a study dedicated to the Scala programming language ecosystem, De Bleser et al. [55] found similar results. Many developers were able to perceive but not identify the smells, even though they recognized a design issue. The most commonly identified test smells were sensitive equality, general fixture, and mystery guest, while no developer was able to correctly identify lazy test.

Corroborating the findings of Kummer et al. [60] and De Bleser et al. [55], the research conducted by Campos et al. [61] provides evidence that developers perceive test smells as being of low or moderate severity in the projects they work on. This study was conducted on six open-source projects, and follow-up interviews were conducted to understand whether test smells affected the test code.

### 4.2.2. The Impact of Test Smells

The findings related to test smells across the industry and academia reveal that despite the existence of these smells, practitioners are often unaware of their names or the impact that such smells have on the code base. To address this gap, the impact of test smells on systems and their diffusion was researched by Bavota et al. [53] in two studies aimed at unveiling the impact on software maintenance. Research indicates that most efforts throughout the software lifecycle are spent on maintenance [62,63]. In the first study, an exploratory analysis of 18 software systems (two industrial and 16 open-source) analyzed the distribution of test smells in source code, the result was as follows:

- High diffusion of smells in both environments was found.
- Open-source projects present fewer test smells compared to industrial ones.

The second study was a controlled experiment involving 20 master students. In the laboratory, students participated in two different sessions. In one session, students worked with a codebase containing code smells, while in the second session, they worked with a codebase free of code smells. The reported results show that test smells negatively affect the comprehension of source code.

Bavota et al. [56] analyzed 25 open-source projects and two industrial projects. The first study aimed to understand the diffusion of test smells in software systems. The results showed that among the 987 analyzed JUnit classes, only 134 were not affected by any test smell, while the majority of the classes (853 in that case) were affected by at least one test smell. Generally, for most types of bad smells, the percentage of instances of these smells is higher in industrial systems, a trend also reported by Bavota et al. [53]. This can potentially be explained by the greater time pressure often found in an industrial context, which may make industrial programmers more prone to bad programming practices.

The same pressure was also observed by Junior et al. [64]. The study gathered data using two methods: a survey with 60 participants and interviews with 50 participants. For the interviews, participants from the survey were invited, and others were contacted via LinkedIn. The yielded results indicate that experienced professionals introduce test smells during their daily programming tasks, even when adhering to their company's standardized practices. One practitioner reported that the project context is sometimes not ideal for practicing TDD, suggesting that there is a rush to deliver the feature. Additionally, experienced practitioners may not produce fewer test smells compared to less experienced ones. The same result was found by Kim et al. [65]. In their study, they analyzed 12 real-world open-source systems and found that experienced developers often do not refactor test smells due to a lack of awareness.

### 4.3. Tooling

Attention to developer tooling is crucial because a significant portion of development time and effort is dedicated to the production and maintenance of test suites. In line with this, researchers have developed a tool aimed at enhancing developers' awareness of test

suite quality, named TestSmellDescriber [63]. The tool generates test case summaries that provide developers with textual aids for understanding the test case. The tool was created based on a theoretical framework that connects code smells with test smells. It comprises three steps to generate the summaries:

- Smell Detection—This phase uses two tools: DECOR (Detection and Correction) and TACO (Textual Analysis for Code Smell Detection).
- Summary Generation—In this phase, the summary generated is implemented using the standard software word usage model.
- Description Augmentation—In this last phase, TestSmellDescriber augments the descriptions with the test method and test class.

The tool adds comments at the method level with recommended actions to fix each detected smell. The outcomes of a study conducted by the authors suggest positive results in assisting developers with improving the quality of the test suite.

Continuing along the path of improving testing practices, Brandt and Zaidman [66] investigated how to design a developer-centric test amplification approach for success with developers. To achieve this goal, they developed a tool named 'Test Cube' and made it available in the JetBrains store https://plugins.jetbrains.com (accessed on 15 February 2024) for developers to try. The research recruited participants from Twitter, contacts in the industry, and previous participants who had shown interest in joining the research. Participants were given time to interact with the tool using a setup arranged by the researchers. This strategy aimed to streamline the process, as HTML is familiar to developers, thus requiring less time to explain the domain. The study recorded interviews with a total of 16 participants, along with their screens. The research results include the following observations:

- Participants attempted to understand the test cases, and many of them renamed the generated test cases. They became annoyed because the generated variables could be inlined.
- Share why newly covered functions are considered important.
- Participants suggested that a progress bar would help them identify when the tool is running.

The technique of amplifying new test cases based on participant feedback is a tool that helps developers discover potential new test cases to increase confidence in the test suite. On the other hand, the manual generation of test cases poses challenges, as depicted by Axelrod [67].

The diffusion of test smells in automatically generated test code was also investigated by Palomba et al. [57]. One noticeable aspect is the percentage of JUnit test classes containing at least one test smell. It was found that 13,791 out of a total of 16,603 JUnit classes (83% of them) are affected by design problems. Particularly, assertion roulette, eager test, and test code duplication are widespread. The results suggest that 83% of the 16,603 JUnit classes are affected by design problems. In their analysis, they found that all test smells had a strong positive correlation with the structural characteristics of the systems, such as size or number of classes. The larger the system, the higher the likelihood that its JUnit classes are affected by test smells. Consequently, this might negatively impact developers in finding faults in the production code.

## 5. Discussion

Test smells are a well-researched topic in the community, and as such, their popularity has been explored from various angles, including natural test smells, automated test smells, and the development of tools to help developers better assess their test suites. In this section, we present the findings and contrast them with the research questions.

### 5.1. RQ1: When Developers Practice TDD, Are There External Influences on Developers to Use TDD?

The nature of software development is complex and requires different roles to work together to achieve the desired outcome. Research has shown that different roles have varying viewpoints in the process, with the improvement of the organization being the ultimate goal. However, an effective testing strategy also takes into account these different roles.

Studies have indicated that automation is crucial for a maturity model seeking to streamline the end-to-end process in the software delivery chain. The higher the level of automation, the greater the maturity. Empirical evidence highlights the challenges that companies face when adopting a DevOps strategy. It is essential to note that testing strategy and automation are at the core of a well-structured software delivery pipeline.

From an academic standpoint, teaching test-driven development (TDD) is challenging. Delivering the best content requires professional experience in the field, as research has shown that students tend to value different aspects of the practice compared to professionals.

Not only academia faces challenges with this strategy, but also companies aiming to adopt TDD to enhance their software delivery. The objective is twofold: to adopt the methodology and achieve business outcomes. Bringing in a champion who already knows the practice can be advantageous for newcomers. Thus, learning occurs simultaneously with practice.

Regarding practice, TDD has been extensively researched by the community. Although the promise of TDD generating better code design lacks empirical evidence, practitioners value the constant feedback it offers. Additionally, practices closely associated with TDD, such as pair programming, have gained popularity through methodologies like eXtreme Programming. This interest in the practice has led to an exploration of what constitutes a good test and the principles to consider when writing tests to ensure their value. The literature provides a set of patterns for building code guided by tests and how to use code for testing.

All in all, practitioners advocating for the incorporation of TDD practice can benefit from the outcomes of this study in the following ways:

- The testing strategy affects the practice of TDD in industrial projects, leading practitioners to drop the practice due to pressure.
- Practitioners should take into account the negative impact that test smells have on code comprehension in the daily practice of writing tests. The body of knowledge offered by academia provides empirical evidence for that.

And for researchers, the following points have room for improvement and warrant empirical investigation:

- When practitioners abandon the TDD practice due to pressure, further investigation is needed to understand the long-term effects it may have.
- Aligning the testing strategy with business outcomes and the various roles leads to an increased maturity level in automation.

### 5.2. RQ2: Does the Practice of TDD Influence the Addition or Maintenance of New Test Cases?

Test smells, on the other hand, are referenced in various types of testing and approaches. There appears to be a discrepancy between academia and the gray literature, as different sources consumed by professional software developers point in another direction. Even the blending of test smells and anti-patterns seems to be in a gray area, yet the understanding of both remains the same.

On one hand, research has shown that despite the vast body of knowledge in academia, practitioners perceive test smells as ranging from low to medium severity in an industrial context. In open-source projects, the evidence indicates less diffusion of test smells. The literature highlights the time pressure imposed on industrial practitioners to deliver features. On the other hand, there is also evidence that test smells negatively impact the comprehensibility of the code.

The research community has also focused on delivering solutions to address test smells, although the TDD practice is not the primary focus in these studies. For example, TestSmellsDescriber offers a command-line tool that adds comments at the method level, with the test smells found and possible fixes. More recently, Test Cube has also been incorporated into the tools available via the JetBrains marketplace. Test Cube offers developers new classes in an attempt to improve the test suite quality. The results gathered by researchers point to a tool that practitioners would use in real industry projects.

To conclude, practitioners can benefit from the following:

- Academia offers a set of tools that practitioners can benefit from to detect test smells and improve the quality of the test suite for Java-based projects.
- The term referred to varies between academia and industry. In general, academia uses the term "test smells", while the industry uses "anti-patterns" to refer to the same subject. Such differences might hinder the adoption and discovery of tools that practitioners can benefit from in industry projects. Using both terms when searching for test suite improvements can enhance the accessibility of results

And researchers should benefit from the following:

- TDD styles in industry projects lack scrutiny regarding the effects the style plays in generating test smells. While TDD styles are mentioned in books targeting practitioners, there is room for improvement in terms of empirical data.
- Testing strategies based on context impact TDD practice. Studies should take into account these settings when exploring the effects of the practice. Similarly, as the previous point mentioned, such context offers an area for exploring the intersection between TDD practice and the generation of test smells.
- Regarding tooling specifically for test smells, academia has a vast body of knowledge, and this study specifically highlights two of them: TestSmellsDescriber and Test Cube. Despite the studies done to validate such tools, there is room for improvement regarding their usage in the wild by practitioners and their effectiveness, focusing on industrial outcomes.
- The tools investigated in this study indicate a greater adherence to the Java ecosystem than other programming languages. This study depicts that, based on the programming language, other aspects of testing come into play. For researchers, other ecosystems such as JavaScript/TypeScript, PHP, Python, and other programming languages should also be targets for tools to detect test smells. Such ecosystems might point to other types of smells.

## 6. Conclusions

Throughout this paper, a systematic literature review was conducted following the PRISMA framework. The first step was to define the research questions, followed by a shallow analysis of the existing SLRs in the literature, leading to the definition of the PIACOC and the identification of Scopus, Web of Science, and Google Scholar as databases of interest. The analysis of 2036 papers across three databases was performed, applying the inclusion and exclusion criteria, and utilizing filter criteria aligned with the mapping questions of the research. In total, 28 papers were selected, and the results were discussed.

The result of this work is broader in the sense that it delves into how software is written through DevOps maturity models, progressing towards a testing strategy that takes into account the different aspects of an organization, ultimately leading to the TDD practice and later on, in the production related to test smells. While test smells are known in academia, anti-patterns are also used in the industry. The community has provided empirical evidence that test smells are harmful for comprehension, and tools have been developed to help practitioners improve their test suites. However, practitioners often rank them as a low to medium priority, even though they have a negative impact on code comprehension.

Despite the research presented, the literature shows a gap in the end-to-end framework to measure how the practice of TDD leads to anti-patterns/test smells. The research outcome indicates that the areas of research are well defined when both subjects are treated separately. Attempts have been made to integrate these two areas, but there is room for further investigation in this area. The work presented here illustrates a gap, and no set of metrics is used to measure if the impact of TDD practice is also the reason for the test smells. It is crucial for researchers and practitioners to collaborate in developing guidelines for implementing an effective testing strategy centered on TDD, aiming to minimize test smells in code bases. This endeavor should encompass the entire spectrum of the software development process, prioritizing a test-first approach. Moreover, conducting assessments beforehand could help teams identify the most optimal strategy. This work establishes a foundation for further investigation and propels progress in the solution space.

**Author Contributions:** Conceptualization, M.M., A.G.-H. and F.J.G.-P.; methodology, M.M., A.G.-H. and F.J.G.-P.; formal analysis, M.M.; investigation, M.M.; resources, M.M.; data curation, M.M., A.G.-H. and F.J.G.-P.; writing—original draft preparation, M.M.; writing—review and editing, M.M.; visualization, M.M.; supervision, A.G.-H. and F.J.G.-P. All authors have read and agreed to the published version of the manuscript.

**Funding:** This research received no external funding.

**Data Availability Statement:** The dataset and scripts used in this systematic literature review is available at https://github.com/marabesi/slr-tdd-anti-patterns-test-smells (accessed on 16 March 2020).

**Conflicts of Interest:** The authors declare no conflicts of interest.

## Abbreviations

The following abbreviations are used in this manuscript:

| | |
|---|---|
| TDD | Test-Driven Development |
| TLD | Test-Last Development |
| BDD | Behavior-Drive Development |
| KLOC | Thousands Of Lines Of Code |
| LOC | Lines Of Code |
| DDD | Domain Driven Design |
| MDD | Model Driven Development |
| PICOC | Population, Intervention, Comparative, Outcomes and Contex |
| CI | Inclusion Criteria |
| CE | Exclusion Criteria |
| Q | Question |
| CSV | Comma Separated Value |
| JSON | JavaScript Object Notation |
| KO | Knockout |
| DevOps | Development and Operations |
| UI | User Interface |
| IT | Information Technology |
| OOP | Object Oriented Programing |
| HTML | Hypertext Markup Language |
| DECOR | Detection and Correction |
| TACO | Textual Analysis for Code Smell Detection |

## Appendix A

**Table A1.** Broader context.

| Title | Production Type |
|---|---|
| An empirical analysis of the distribution of unit test smells and their impact on software maintenance [53] | Conference Paper (P) |
| The secret life of test smells-an empirical study on test smell evolution and maintenance [65] | Conference Paper (P) |
| Test Driven Development: Advancing Knowledge by Conjecture and Confirmation [43] | Conference Paper (P) |
| Defining suitable testing levels, methods and practices for an agile web application project [36] | Thesis (T) |
| DevOps Capability Assessment in a Software Development Team [38] | Thesis (T) |
| Knowledge and mindset in software development–how developers, testers, technical writers and managers differ  a survey [33] | Conference Paper (P) |
| xUnit test patterns: Refactoring test code [7] | Book (B) |

**Table A2.** TDD/Teaching.

| Title | Production Type |
|---|---|
| The role of unit testing in training [41] | Conference Paper (P) |
| How Students Unit Test: Perceptions, Practices, and Pitfalls [46] | Conference Paper (P) |

**Table A3.** Anti-patterns/test smells/broader context.

| Title | Production Type |
|---|---|
| How are test smells treated in the wild? A tale of two empirical studies [64] | Conference Paper (P) |
| Developers perception on the severity of test smells: an empirical study [61] | Conference Paper (P) |

**Table A4.** TDD overall/testing practices.

| Title | Production Type |
|---|---|
| How the practice of TDD influences class design in object-oriented systems: Patterns of unit tests feedback [4] | Conference Paper (P) |
| Developer-centric test amplification: The interplay between automatic generation human exploration [66] | Conference Paper (P) |
| Adding More Tests [67] | Book Chapter (BC) |
| Predicting software test effort in iterative development using a dynamic Bayesian network [42] | Conference Paper (P) |
| Rationales and Approaches for Automated Testing of JavaScript and Standard ML [50] | Conference Paper (P) |
| Developer testing: Building quality into software [39] | Book (B) |
| Sustainable software development through overlapping pair rotation [35] | Conference Paper (P) |
| Learning effective test driven development—Software development projects in an energy company [47] | Conference Paper (P) |

**Table A5.** Anti-patterns/test smells.

| Title | Production Type |
|---|---|
| How Good Are My Tests? [49] | Conference Paper (P) |
| Categorising test smells [60] | Thesis (T) |
| Smells in software test code: A survey of knowledge in industry and academia [58] | Conference Paper (P) |
| Enhancing developers' awareness on test suites' quality with test smell summaries [63] | Thesis (T) |
| Hunting for smells in natural language tests [54] | Conference Paper (P) |
| Smells in system user interactive tests [68] | Conference Paper (P) |
| Assessing diffusion and perception of test smells in scala projects [55] | Conference Paper (P) |
| Are test smells really harmful? An empirical study [56] | Conference Paper (P) |
| On the diffusion of test smells in automatically generated test code: An empirical study [57] | Conference Paper (P) |

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
