# Peer review of "Exploring the Connection between the TDD Practice and Test Smells—A Systematic Literature Review"

_computers, doi:10.3390/computers13030079_

Round 1

Reviewer 1 Report

Comments and Suggestions for Authors

(1) The author’s research motivation is unclear as a systematic review.

(2) What specific examples of TDD exist?

(3) The author's purpose and contribution should be concentrated in the first chapter.

(4) The reviewers consider that related work should be summarized in a new Chapter 2.

(5) This article has too many small paragraphs, and the connection between the core points is unclear.

That's all. Thanks.

Reviewer 2 Report

Comments and Suggestions for Authors

The subject of the article is relevant. The research gap needs more focus on the subject of the study although it is clear that this work is justified. There are several decisions that need to be better justified. The exploration of the results is too descriptive and needs more organization.

Improvement suggestions:

1. The research gap lacks focus. It contains too much information regarding the benefits of TDD which is not the main goal of this study. The intersection between TDD and test smells.

2. Introduction section doesn’t explore the concept of tests smells and how agile could be used on it.

3. Research questions proposed in this study should be aligned with the research gap. Currently, it was not done.

4. Google Scholar is often cited as a source for gray literature, especially in the search methodology for systematic reviews. How did you manage this issue?

5. The number of studies identified in google scholar is dependent of cache mechanisms and geographic location of authors. How did you manage this issue?

6. In the screening phase the authors identified 1691 records and have excluded 1419 excluded. Based on which criteria? Was it based on a manual or automatic process?

7. The second phase of the screening phase is irrelevant because all reports were retrieval.

8. Authors excluded 244 studies based on the criteria for quality evaluation. Why < 2 and not <3, considering that the maximum value is 6?

9. The categories presented in Table 4 are not properly defined and explored.

10. I really didn’t understand the organization of the results section. The topic addressed in 3.1 should also be better addressed considering their relevance. The exploration of the topics regarding their contributions should also be better explored.

11. I would expect to have more information regarding the research teams, universities and the connection between some topics. This could be done using software as VosViewer? Did you use any software for explore the results?

12. It would be fundamental to establish a research agenda in this field. Currently, it was not done.

13. Practical contributions of this study should be explored.

Comments on the Quality of English Language

Proofreading is highly relevant. 

Round 2

Reviewer 1 Report

Comments and Suggestions for Authors

The reviewer is pleased that the authors have made most of the revisions to this article. Here are the new comments.

(1) The text in Figures 1-6 and 8 is very small. The author should improve them to make them clear to readers.

(2) For TDD-related references, what are the characteristics of different systematic literature libraries as sources? For example, Springer, IEEE eXplore.

(3) How many studies in the data set investigated by the author relied on manual data collection and analysis? Which ones take advantage of the latest technology? For example, generative AI, etc.

(4) What possible connections and applications does the contribution of this study have for future related work?

That's all. Thanks.

Reviewer 2 Report

Comments and Suggestions for Authors

The research gap needs to be improved. The objectives of the work are clear, but the need for a systematic review in the field is not. What other systematic reviews are there? What is the unique contribution of this review?

The dates when the systemic review was carried out should be specified.

I also recommend justifying the search keywords in the databases, as this is a fundamental element of the process.

Comments on the Quality of English Language

it is ok.
